# ArtiFact: A Large-Scale Multi-Modal Cultural Heritage Dataset

Luciano Duarte
BIFOLD & TU Berlin
duarte.castineira@tu-berlin.de

Olga Ovcharenko
BIFOLD & TU Berlin
ovcharenko@tu-berlin.de

Sebastian Schelter
BIFOLD & TU Berlin
schelter@tu-berlin.de

## ABSTRACT

Multi-modal data management has emerged as a central research topic in the database community, spanning data integration, semantic query processing, and data quality assessment. Despite this growing interest, the community lacks large-scale, real-world datasets combining tables, text, and images. We present ArtiFact, a multi-modal cultural heritage dataset of 651,045 museum records collected from the Metropolitan Museum of Art, the Art Institute of Chicago, and the Rijksmuseum. We demonstrate the utility of ArtiFact through two downstream tasks. For cross-modal error detection, we introduce a curated taxonomy of seven error categories injected into 130,209 records and show that reliably detecting subtle domain-specific errors such as material anachronisms and temporal shifts remain an open challenge. For semantic query processing, we show that current systems struggle with queries involving cultural proximity, ambiguous object types, and historically contingent terminology. Our results position ArtiFact as a challenging benchmark for multi-modal data management research.

**VLDB Workshop Reference Format:**
Luciano Duarte, Olga Ovcharenko, and Sebastian Schelter. ArtiFact: A Large-Scale Multi-Modal Cultural Heritage Dataset. VLDB 2026 Workshop: NOVAS.

**VLDB Workshop Artifact Availability:**
The source code, data, and/or other artifacts have been made available at https://olgaovcharenko.github.io/ArtiFact/.

## 1 INTRODUCTION

Multi-modal data management has emerged as a central research frontier in the database community, spanning data integration [41], semantic query processing [13, 18, 19, 28, 30], entity resolution [45], and data quality assessment [29]. As collections increasingly pair structured records with unstructured text and images, fundamental questions arise about how to store, index, query, and clean data across modalities. However, despite growing interest in data-centric AI and omni-modal models [12], little attention has been paid to the construction of large-scale and well-curated multi-modal datasets spanning tabular, textual, and image data. Many existing benchmarks implicitly assume that associated metadata and images are correct and consistent [25], even though real-world collections often contain noisy annotations, ambiguous terminology, missing information, and historical inconsistencies.

**Lack of datasets for the cultural heritage domain.** This gap is particularly pronounced in the cultural heritage domain. Museums and archives worldwide are rapidly digitizing collections and publishing open-access records containing artwork images and catalog descriptions containing metadata, materials, artist attributions, and geographic information. These collections represent a uniquely rich source of multi-modal knowledge collected by domain experts spanning centuries of cultural heritage. At the same time, these collected records originate from heterogeneous metadata standards, evolving institutional practices, and manual documentation workflows, making large-scale normalization, interoperability, and validation inherently challenging [44]. Existing large-scale collections such as OmniArt [39], SemArt [5] (private), SILKNOW [33], and TimeTravel [7] provide valuable resources for retrieval, classification, and multi-modal reasoning tasks, leaving data quality and metadata correctness unexamined. Similarly, cross-modal error detection benchmarks such as MERIT [29] and MMIR [46] demonstrate the importance of multi-modal data quality evaluation, yet remain restricted to domains such as e-Commerce and web data. Open source large-scale datasets for multi-modal data management remain scarce, especially in the cultural domain [6].

**ArtiFact dataset.** To address this gap, we introduce ArtiFact, a large-scale multi-modal cultural heritage dataset consisting of 651,045 artwork records paired with public-domain images and normalized structured metadata. The dataset aggregates open-access collections from three major institutions: the Metropolitan Museum of Art (MET) [40], the Art Institute of Chicago (AIC) [2], and the Rijksmuseum in Amsterdam [35]. Beyond aggregating records, ArtiFact provides a unified preprocessing and normalization pipeline that standardizes heterogeneous museum metadata through rule-based processing and LLM-assisted semantic parsing, as shown in Figure 1. Additionally, ArtiFact supports a broad range of downstream multi-modal tasks in cultural heritage collections, including semantic query processing, multi-modal data quality assessment, and cross-modal error detection [29]. To the latter, the dataset incorporates a curated taxonomy of synthetically injected errors, informed by domain experts, spanning temporal, geographic, cultural, material, spatial, identity, and visual dimensions. In this paper, we showcase two downstream tasks: cross-modal error detection and semantic query processing.

**Contributions.** Our contributions are as follows.

- A *dataset* of 651,045 museum objects paired with tables, images, and text, collected from three cultural heritage institutions. The dataset is available at https://olgaovcharenko.github.io/ArtiFact/.
- A *unified data cleaning pipeline* to parse and normalize heterogeneous museum data (Section 3).
- An evaluation of ArtiFact on two *cross-modal error detection* and *semantic query processing* tasks, showcasing the dataset as a testbed for multi-modal data management research. For error

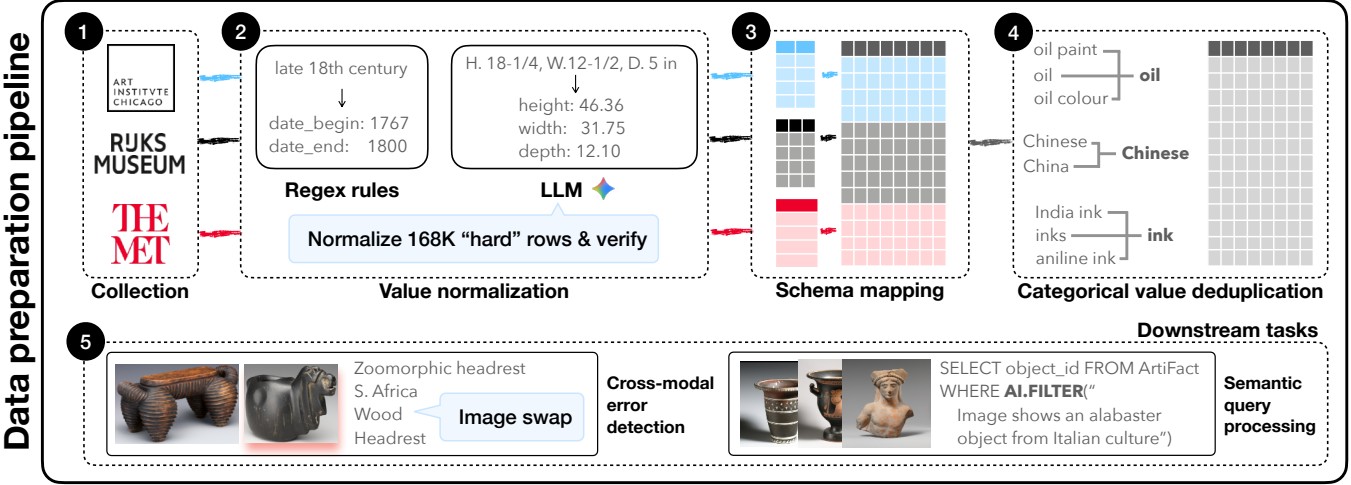

**Figure 1: Overview of the dataset generation process for `ArtiFact`: We run an ETL pipeline to ❶ collect, ❷ structure, ❸ normalize, and ❹ deduplicate data from three cultural heritage institutions; the preprocessed dataset can be used for various ❺ downstream tasks including cross-modal error detection and semantic query processing.**

detection, we develop a curated error taxonomy informed by museum curators, and highlight the difficulty of each error type. For semantic query processing, we show that systems have issues in dealing with cultural biases and historical nuances (Section 4).

## 2 RELATED WORK

We discuss related work on multi-modal cultural heritage datasets, data cleaning, and multi-modal data preparation benchmarks.

**Multi-modal cultural heritage datasets.** SILKNOW [33], OmniArt [39], SemArt [5], and TimeTravel [7] aggregate large collections for classification, retrieval, or cultural question answering. ViMUL-Bench [37], ALM-Bench [42], and DRISHTIKON [22] evaluate LLM question-answering across cultures and languages. None of these datasets jointly provide tabular, image, and textual data.

**Multi-modal data preparation.** Tabular error detection has a long history in the data management community [1, 3, 15, 36]. HoloClean and HoloDetect [34] apply probabilistic inference and few-shot learning, Raha [21] and Baran [20] ensemble heterogeneous detectors and repair rules, ActiveClean [16] integrates active learning, and Deequ [36] automates declarative data quality validation at scale. However, all tabular methods are confined to within- or inter-table inconsistencies and are not designed for multi-modal data. Another line of work recasts cleaning as label error detection on a target column: Cleanlab [23, 24] estimates noisy label probabilities via confident learning, Jäger and Biessmann 2024 calibrate uncertainty via conformal learning, methods leveraging Data Shapley values [8, 14, 43] score per-sample influence, and LEMoN [48] contrasts CLIP [31] image and text neighborhoods. Label error detection methods assume a single noisy label per record/label, overlook multi-column errors, and require a separately trained model per target column. LLM- and VLM-based methods such as FineMatch [9], VDC [50], and DataVinci [38] demonstrate that generative reasoning can flag text-image mismatches. MERIT [29] formalizes cross-modal error detection for e-Commerce data and shows that it is beneficial to use multi-modal data for the error

detection. MMIR [46] injects five categories of semantic errors into 534 layout-rich web artifacts (pages, slides, posters) and reports that even proprietary models remain vulnerable to inconsistencies confined to a single element within a complex layout. C3 [49] uses LLMs to evaluate completeness and consistency in cultural-heritage records. MMMU-Pro [47] assesses multi-modal models' understanding and reasoning capabilities.

## 3 THE ARTIFACT DATASET

We discuss the data generation process for `ArtiFact`, see Figure 1.

### 3.1 Data Preprocessing & Normalization

**Data sources.** The `ArtiFact` dataset aggregates open-access records from three major institutions: the Metropolitan Museum of Art (MET) via its REST API across all 19 curatorial departments [40]; the Art Institute of Chicago (AIC) [2] via its open-access JSON-LD files [4], using the International Image Interoperability Framework [10]; and the Rijksmuseum [35] via the Open Archives Initiative Protocol for Metadata Harvesting [17] and recursive parsing of JSON-LD linked data [4]. Only records with public-domain image URLs are retained, yielding 651,045 objects. The majority of records are in English, with a small subset in Dutch.

*Data preparation.* Before integrating the data from three institutions into a single dataset, records from each institution are preprocessed independently. To ensure consistency, global transformations are applied prior to parsing individual fields. We prepend institutional prefixes (e.g., MET_) to all object_IDs to prevent collisions.

*Rule-based value normalization.* First, we normalize all dates into a common format by removing non-ASCII symbols, standardizing terms (e.g., *B.C.*, *B.C.E.* to *BCE*), parsing textual dates to absolute bounds (e.g., *18th century* to *1701-1800*, *late 18th century* to *1767-1800*), and validating time spans. Second, we process the textual dimensions of each cultural heritage object by translating all units to centimeters/grams, restructuring single- and multi-component objects into a standardized textual format (e.g., *10 1/4 by 7 in, weight*

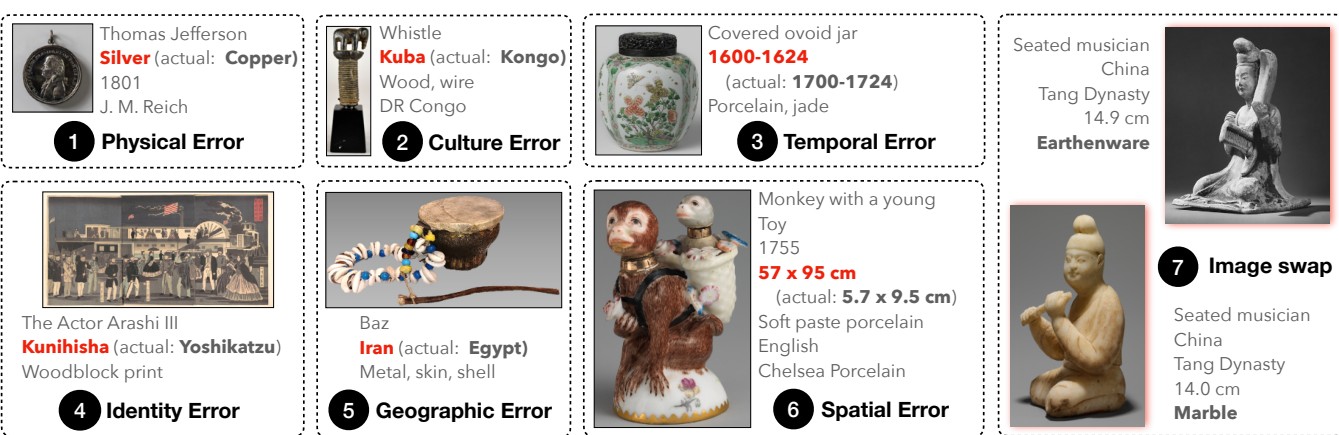

**Figure 2: We inject seven error types into 130,209 data records to create evaluation data for the cross-modal error detection task.**

*12.5 kg* to *height: 26.0 cm, width: 17.8 cm, weight: 12500.0 g*). Third, for the AIC dataset, we split the artist information into five possible fields: artist name, role, nationality, and dates of life (e.g., *Francisco Domingo y Marqués (Spanish, 1842-1920)* to *name: Francisco Domingo y Marqués, nationality: Spanish, dates: 1842-1920*). Fourth, we extract structured arrays of materials and techniques from free text descriptions and map synonyms and typographical errors to a single canonical term via manually developed reference dictionaries of over 1,700 terms (e.g., *agouache* to *gouache*).

*LLM-based value normalization.* We process around 168,000 records with Google's Gemini 2.5 Flash Lite to structure the parts of the data that could not successfully be handled via rule-based parsing (e.g., *albumen silver print on a paper support (hand-colored)* to *materials: paper, techniques: albumen silver print, hand colored* or complex dates). We develop attribute-specific prompts and leverage Chain-of-Thought prompting to improve the outputs.

*Schema mapping.* We consolidate the data collected from the three museums into a single 24-column schema with the following attributes: identifiers (object id, name, title, description, subjects, inscriptions, image URL), temporal (reign, dates, period, dynasty), physical (materials, techniques, dimensions), cultural/geographic (culture, location), artist (artist name, nationality, role, dates).

*Categorical value deduplication.* We map different institutional labels to a shared vocabulary. Material, technique, and object names are embedded with `all-MiniLM-L6-v2`, a sentence Transformer embedding model that is used for clustering and semantic search. The candidate pairs are classified by an LLM into *synonyms*, *parent-child*, or *distinct* to prevent erroneous merging of semantically close concepts (e.g., sword vs. dagger). True synonyms and hierarchical relationships are manually integrated via pre-built dictionaries. Artist deduplication is framed as entity matching: two records are merged if and only if all five cleaned attributes (name, role, nationality, birth and death years) match. We manually map name variations (e.g., *Utagawa Hiroshige* and *Utagawa Hiroshige (I)*) to unique individuals to avoid erroneous merges or duplicates. The complex culture attribute strings (e.g., *Italian, Venice*) are parsed with an LLM to separate cultural descriptors from geographic references, retaining sub-cultures and artistic periods within the culture field while routing specific locations to the location column.

## 4 DOWNSTREAM TASKS

We evaluate `ArtiFact` on two downstream tasks (cross-modal error detection and semantic query processing) to characterize the dataset's difficulty and demonstrate its applicability for multi-modal data management research. Our aim is to give future users empirical guidance for scoping experiments and selecting relevant subsets.

### 4.1 Cross-Modal Error Detection

As multi-modal datasets grow in scale and importance, ensuring their correctness becomes a central data management concern. Cross-modal error detection [29] aims to identify records where tabular data, text, and images are inconsistent. Error **7** in Figure 2 illustrates the problem with two similar museum records depicting a 7th-century Chinese "Seated Musician", one made of marble [27] and the other of earthenware [26]. The images are swapped, but the tabular data is considered clean. `ArtiFact` provides a controlled, realistic setting to study these challenges at scale.

**Error taxonomy & injection.** To develop a realistic error taxonomy, we consulted curators at the MET [40], AIC [2], and Rijksmuseum [35], who named wrong image assignments, typographic errors, and field swaps as the most prevalent error types. We further collect dataset statistics, including temporal timelines, geographic distributions across materials, techniques, and artists, as well as co-occurrence patterns, and use these to cluster artists into cohorts by period and style, and to group cultures along geographic and temporal dimensions. We define seven error categories with nineteen subcategories, see examples in Figure 2. **1** Physical errors inject material or technique anachronisms using the pre-computed statistics (co-occurrences, date/place/nationality patterns for material/technique timelines, geographic distributions, cross-field correlations, artist and culture clusters and adjacency), or swaps between visually similar materials that frequently co-occur with the object type (20,484 data records). **2** Culture errors swap labels between adjacent cultures or within the same continent to ensure the error is plausible but factually incorrect (10,780 data records). **3** Temporal errors shift the creation range by a random historical offset of ±100, 200, or 300 years (8,992 data records). **4** Identity errors simulate professional attribution challenges, from random swaps to entity swaps between artists who share nationality, historical

**Table 1: Cross-modal error detection with Gemini 3 Flash for 200 samples per each error subtype.**

| Category | Error subtype | $\mathcal{P}$ | $\mathcal{R}$ | $\mathcal{F}1$ |
|---|---|---|---|---|
| **Physical** | Technique anachronism | 0.99 ± 0.01 | 0.71 ± 0.01 | 0.83 ± 0.01 |
| | Technique interchange | 0.93 ± 0.01 | 0.67 ± 0.01 | 0.78 ± 0.01 |
| | Material anachronism | 0.70 ± 0.00 | 0.50 ± 0.00 | 0.58 ± 0.00 |
| | Material interchange | 0.79 ± 0.00 | 0.56 ± 0.00 | 0.66 ± 0.00 |
| **Culture** | Culture (continent swap) | 1.00 ± 0.00 | 0.81 ± 0.02 | 0.89 ± 0.01 |
| | Culture (tight adjacency) | 0.82 ± 0.00 | 0.66 ± 0.01 | 0.73 ± 0.00 |
| **Temporal** | Century shift | 0.52 ± 0.01 | 0.80 ± 0.00 | 0.63 ± 0.01 |
| **Identity** | Artist (random) | 0.84 ± 0.00 | 0.92 ± 0.01 | 0.88 ± 0.00 |
| | Artist (era) | 0.70 ± 0.00 | 0.76 ± 0.01 | 0.73 ± 0.01 |
| | Artist (medium) | 0.76 ± 0.01 | 0.83 ± 0.01 | 0.79 ± 0.00 |
| | Artist (specialization) | 0.73 ± 0.01 | 0.80 ± 0.03 | 0.76 ± 0.02 |
| **Geographic** | Place (city level) | 0.70 ± 0.00 | 0.56 ± 0.01 | 0.62 ± 0.01 |
| | Place (country level) | 0.87 ± 0.01 | 0.69 ± 0.02 | 0.77 ± 0.02 |
| **Spatial** | 10x scale error | 0.94 ± 0.00 | 0.99 ± 0.00 | 0.96 ± 0.00 |
| | Aspect ratio swap | 0.51 ± 0.00 | 0.54 ± 0.00 | 0.53 ± 0.00 |
| **Visual** | Embedding swap | 0.80 ± 0.02 | 0.68 ± 0.02 | 0.73 ± 0.02 |
| | Image swap (easy) | 1.00 ± 0.00 | 0.90 ± 0.01 | 0.95 ± 0.01 |
| | Image swap (medium) | 0.99 ± 0.01 | 0.85 ± 0.01 | 0.91 ± 0.01 |
| | Image swap (hard) | 0.96 ± 0.01 | 0.82 ± 0.01 | 0.89 ± 0.01 |
| **No error** | Is the record clean? | 0.23 ± 0.01 | 0.57 ± 0.01 | 0.32 ± 0.01 |

era, and primary specialization (26,952 data records). ❺ Geographic errors exchange locations between neighboring countries while maintaining a lexical overlap constraint to prevent swapping visually identical labels (13,476 data records). ❻ Spatial errors scale the dimension units by 10 and swap the aspect ratios (15,837 data records). ❼ For visual errors, we swap images leveraging CLIP [32] embeddings to identify visual twins. We create adversarial pairs where images are swapped between visually similar records from different contexts (33,688 data records).

**Discussion.** To show the difficulty of `ArtiFact` and provide future users with a reference point, we establish a simple baseline using Gemini-3-Flash (with default temperature) on a representative sample of 200 records per error subtype and 200 clean records (4,000 records total). The results in Table 1 reveal a clear difficulty spectrum across error types, which we consider the primary contribution of this evaluation. Visually or semantically salient errors are reliably detected: image swaps, 10x scale errors, and continent-level culture swaps are recognized with high performance. In contrast, subtle domain-specific inconsistencies like aspect-ratio swaps, material anachronisms, and clean records remain challenging. We observe systematic misclassification patterns, for example, a 19th century Qing dynasty temporal error was deemed historically plausible, and aspect-ratio errors were misattributed to artist inconsistencies suggesting that domain knowledge and multi-modal reasoning are critical for cross-modal detection on `ArtiFact`. Our findings give future users a practical reference for constructing targeted evaluation samples by difficulty/error type.

## 4.2 Semantic Query Processing

Semantic query processing — the execution of LLM-powered SQL queries over multi-modal data — has emerged as a central challenge for the database community [18]. Evaluating such systems

**Table 2: Semantic query processing with Gemini 3 Flash.**

| Query | $\mathcal{F}1$ score | |
|---|---|---|
| | Palimpzest | LOTUS |
| Q1: Chelsea porcelain vases | 0.69 ± 0.01 | **0.79** ± 0.01 |
| Q2: British alabaster pieces | 0.71 ± 0.02 | **0.85** ± 0.02 |
| Q3: Ukrainian object classification | **0.58** ± 0.00 | **0.58** ± 0.01 |
| Q4: Chinese swords | 0.49 ± 0.01 | **0.58** ± 0.01 |
| Q5: Indian albumen silver print | **0.57** ± 0.01 | 0.39 ± 0.01 |

requires datasets that combine structured and unstructured data, cover diverse semantic operators, and provide reliable ground truth. Existing benchmarks such as SemBench [18] provide a strong foundation, but remain limited to a small number of domains and LLM-contaminated datasets. `ArtiFact` complements these efforts by offering a large-scale, domain-specific, and visually rich collection where semantic queries cover tables, textual descriptions with domain-specific terminology, and images with cultural context.

**Queries.** To showcase `ArtiFact` for semantic query processing, we construct five semantic queries, implemented in LOTUS [30] and Palimpzest [19]. Ground truth labels are derived from the original data and are hidden from the semantic query engines. The benchmark queries combine relational predicates over structured data with semantic operators over textual and visual data. Q1 finds porcelain vases produced by the Chelsea Porcelain Manufactory. Q2 retrieves British alabaster objects. Q3 classifies Ukrainian museum objects to a fixed set of categories. Q4 retrieves swords from China. Q5 finds Indian objects produced via albumen silver printing.

**Discussion.** Table 2 shows the performance semantic query processing systems on cultural heritage data. Object appearance ambiguity is a consistent source of error: historical vases with figurative shapes cause many false positives in Q1, and Q3 missclassifies Ukrainian dress ornaments as medallions, likely due to the visual gap between their historical forms and the modern representations prevalent in LLM training data. Geographic and cultural proximity compounds these errors: Q2 struggles to separate British alabaster objects from visually similar Dutch and German artifacts, and Q4 frequently confuses Chinese swords with Japanese, Tibetan, and Hittite objects. Finally, Q5 shows the difficulty of differentiating between different techniques, where dyeing, weaving, and washing are conflated with albumen silver printing. Our results suggest that without domain-specific knowledge, semantic query systems cannot reliably distinguish between neighboring cultures, resolve historically contingent material and technique terminology, or account for how object appearances evolve across time and geography.

## 5 CONCLUSION

We presented `ArtiFact`, a large-scale multi-modal dataset of 651,045 cultural heritage records combining tabular, data, textual, and image data from three major museums, together with a unified ETL pipeline for cross-institutional normalization. Our evaluation on cross-modal error detection and semantic query processing reveals that current systems handle visually salient inconsistencies well but fail on subtle domain-specific nuances requiring cultural and historical knowledge. We hope that `ArtiFact` will stimulate further research on multi-modal data management in cultural heritage domain, and in particular on the cultural and historical biases.

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
