# OpenReview forum: "ArtiFact: A Large-Scale Multi-Modal Cultural Heritage Dataset"
_VLDB.org/2026/Workshop/NOVAS — NOVAS 2026_

### Official Review · Reviewer_yu9W · 2026-07-07

**Confidence:** 3

**Improvement Opportunities:**

a) I appreciate the huge effort the authors have put in collecting the data, consulting with experts, creating a good ETL, putting it in HF etc. However, there is still more work to be done to make the dataset more useful. For e.g. all evaluation is on Gemini 3 Flash and Gemini 2.5 Flash Lite. It is a frontier model but it is not clear if the errors/difficulties are generic or specific to Gemini
b) I understand the rationale for injected errors but it needs some more care. The 7 types of injected errors are indeed (reasonably) realistic. It is not clear to me how much semantically plausible each error are. Having some expert review samples from each subtype would be useful. I worry that some injected errors could be historically anachronous that a semi decent LLM can easily find it out.
c) The semantic query benchmark are simple/not diverse and focuses on small number of queries.
d) The use of 3 museums makes it heterogeneous. However, there is a potential issue of bias as these artifacts are skewed towards western art. I am sure the datasets have artifacts from other cultures (e.g. Asian/African etc). Even if you do not make it more diverse, mentioning the "origin" could make the dataset bias/imbalance clearer

**Minor Comments:**

a) Please include some statistics table either in the paper or in the dataset website.

**Short Summary:**

The paper introduces ArtiFact, a large-scale multi-modal cultural heritage dataset. It has 600K+ museum object records from the 3 museums. Each record has multimodal data: structured metadata, text, and public-domain images. The authors propose an ETL pipeline for common data integration/analysis tasks. The authors use the dataset for two tasks: cross-modal error detection and semantic query processing. The claim is that ArtiFact provides a challenging benchmark for multi-modal data management.

**Strong Points:**

a) I did some googling. It seems that ArtiFact is larger if not comparable to other cultural heritage benchmark datasets. The metadata is heterogeneous (from different continents). The tasks are also genuinely multimodal and involves all 3 modalities. The authors convincingly argue that the dataset is challenging due to various factors such as contingency, ambiguity, evolving nature, schema heterogeneity etc
b) The ETL pipeline is well thought out.
c) Creating/augmenting the error taxonomy in consultation with experts makes the dataset valuable. It is more thoughtful and fine grained than a typical multi-modal dataset.
d) I especially liked the semantic query processing use case. It opens lot of interesting research in semantic query processing with DB community specific optimizations.

---

### Official Review · Reviewer_8fr8 · 2026-07-10

**Confidence:** 4

**Improvement Opportunities:**

O1: The title could better reflect the paper's actual contributions. Before reading the paper, I expected a more generic dataset paper, while the work is really centered around benchmarking important multi-modal data management problems.

O2: The notion of "multi-modal" should be defined more formally. While the paper combines tables, text, and images, it would be useful to explicitly define what multi-modal means in this context and clarify how alignments between modalities are established. In particular, it is not entirely clear whether the benchmark primarily evaluates cross-modal reasoning or also reasoning within and across combinations of modalities.

O3: The benchmark could potentially support additional data management tasks. The preprocessing and normalization pipeline already performs a variety of operations such as normalization and deduplication. These transformations are themselves important data management challenges and could potentially be extended benchmark tasks. Assuming that the processing logs are available, extending ArtiFact to cover such tasks may require relatively limited additional effort while substantially increasing the benchmark's impact.

O4: The evaluation could better justify the choice of baselines. In particular, it is not entirely clear why relevant methods such as MERIT and MMIR, which are discussed extensively in the related work, are not included in the empirical evaluation.

**Minor Comments:**

A short discussion on benchmark limitations and potential dataset biases would be useful.

**Short Summary:**

This paper introduces ArtiFact, a large-scale multi-modal cultural heritage dataset containing over 650K museum records combining tables, text, and images collected from three major museums. In addition to providing a data preperation pipeline, the authors demonstrate the dataset through two downstream tasks: error detection and semantic query processing.

**Strong Points:**

S1: A multi-modal benchmark is a valuable contribution to the data management community and will be of interest to many researchers in the field.

S2: The paper is well written, easy to follow, and the motivations are compelling.

S3: The paper provides preliminary evaluation through two downstream applications.

---

### Official Review · Reviewer_x81C · 2026-07-14

**Confidence:** 4

**Improvement Opportunities:**

1. ArtiFact inherits noise from the source museum records and introduces additional processing through rules, LLM parsing, embedding-based candidate generation, and manually curated merges. In particular, approximately 168K records are structured using Gemini, but the paper does not report field-level precision/recall, parsing-failure rates, or agreement from human verification. Systematic errors in the nominally clean records would affect both the labels in the synthetic-error benchmark and the ground truth for semantic queries. The paper should include a stratified audit by museum, field, and processing path.

2. The taxonomy is informed by curator consultation, which is a strength, but the paper does not quantify whether independent experts judge the injected records to be plausible yet incorrect, or whether the subtype distribution and difficulty reflect errors observed in museum databases. Table 1 evaluates approximately balanced samples of 200 examples per subtype plus 200 clean examples, whereas the prevalence of actual errors is likely much lower.

**Minor Comments:**

The paper should clarify how many repetitions, random sources, and sampling procedures underlie the “±” values in Table 1. This is particularly important because the baseline uses the default temperature. The release would benefit from a datasheet that consolidates the licensing terms for all three institutions, snapshot dates, handling of dead image URLs, train/validation/test splits, and a compatibility policy for future versions. The discussion of “LLM-contaminated datasets” should also be more cautious, since public museum metadata may itself have appeared in foundation-model training corpora.

**Short Summary:**

This paper releases ArtiFact, a multimodal cultural-heritage dataset containing 651,045 collection records with public-domain images. The authors map heterogeneous metadata from the three institutions into a unified 24-column schema using rules, manually curated dictionaries, embedding- and LLM-assisted deduplication, and Gemini 2.5 Flash Lite for parsing. The paper also constructs two downstream tasks. The first injects seven broad categories and 19 subtypes of errors into 130,209 records for cross-modal error detection. The second evaluates five semantic queries over structured fields, text, and images using LOTUS and Palimpzest.

**Strong Points:**

1. The dataset fills a clear gap in realistic multimodal resources for cultural-heritage data management.

2. The data preparation exhibits substantial engineering depth rather than simple aggregation.

3. The error taxonomy and initial difficulty profile provide a useful foundation for future research.

---

### Decision · Program_Chairs · 2026-07-16

**Decision:**

Accept

**Comment:**

The reviewers agree that ArtiFact is a valuable large-scale multimodal benchmark that fills an important gap in cultural-heritage data management. They highlight the substantial engineering effort behind the unified dataset, the thoughtful expert-informed error taxonomy, and the promising downstream tasks in cross-modal error detection and semantic query processing. The dataset provides a strong foundation for research on multimodal data integration, cleaning, and querying, and we hope this work sparks interesting discussion and future research at the workshop.